# 3D Printing of Pediatric Medication: The End of Bad Tasting Oral Liquids?—A Scoping Review

**DOI:** 10.3390/pharmaceutics14020416

**Published:** 2022-02-14

**Authors:** Iris Lafeber, Elisabeth J. Ruijgrok, Henk-Jan Guchelaar, Kirsten J. M. Schimmel

**Affiliations:** 1Department of Clinical Pharmacy and Toxicology, Leiden University Medical Center, Albinusdreef 2, 2333 ZA Leiden, The Netherlands; i.lafeber@lumc.nl (I.L.); h.j.guchelaar@lumc.nl (H.-J.G.); 2Department of Hospital Pharmacy, Erasmus MC—Sophia Children’s Hospital, University Medical Center, Dr. Molewaterplein 40, 3015 GD Rotterdam, The Netherlands; e.j.ruijgrok@erasmusmc.nl

**Keywords:** 3D printing, pediatrics, compounded drug formulations, drug formulation development, personalized medicine, pharmaceutical technology

## Abstract

3D printing of pediatric-centered drug formulations can provide suitable alternatives to current treatment options, though further research is still warranted for successful clinical implementation of these innovative drug products. Extensive research has been conducted on the compliance of 3D-printed drug products to a pediatric quality target product profile. The 3D-printed tablets were of particular interest in providing superior dosing and release profile similarity compared to conventional drug manipulation and compounding methods, such as oral liquids. In the future, acceptance of 3D-printed tablets in the pediatric patient population might be better than current treatments due to improved palatability. Further research should focus on expanding clinical knowledge, providing regulatory guidance and expansion of the product range, including dosage form possibilities. Moreover, it should enable the use of diverse good manufacturing practice (GMP)-ready 3D printing techniques for the production of various drug products for the pediatric patient population.

## 1. Introduction

Pediatric drug formulations require an accurate and flexible dosing strategy, child-friendly dosage forms and specific suitability of the excipients for children. Therefore, the development of pediatric drug formulations is challenging and complex. Furthermore, it is inhibited by a lack of economic incentive [1].

In Europe, before the introduction of European Union (EU) legislation on medicines for children in 2006, pediatric-centered treatment was very limited [2]. Children were often prescribed medicines that were authorized for adults, but that had not been tested or adapted for the pediatric population. These medicines were, therefore, used off-label and could be unsuitable for children in terms of dose, dosage form, and used excipients [3]. In the years before the pediatric regulation [4], between 2004 and 2006, only 30 new medicines and indications were authorized specifically for pediatric use. Between 2014 and 2016, 10 years later, this was increased to 74 new medicines and indications [2]. While the number of new medicines and indications for pediatric use has increased, it is still not enough to meet the need for pediatric-centered drug formulations.

In practice, it is often still necessary to either adjust a marketed drug before it can be administered to a child, or to prepare an extemporaneous formulation (e.g., oral liquid) by the pharmacist. Manipulation of marketed drugs, e.g., crushing or splitting tablets or diluting a solution, often holds two specific reasons, namely, to acquire the needed dosage strength, or to adapt an unsuitable dosage form, i.e., poor swallowability or palatability of the dosage form. A combination of these reasons also occurs [5,6,7,8].

Studies conducted in a children’s hospital in the Netherlands and pediatric wards in a German hospital show that manipulation was necessary for 37% of oral medicine prior to administration [7,8]. Similar studies conducted in Norway and Sweden found a lower manipulation prevalence of 17% and 15.5%, respectively, for orally administered pediatric medicines [5,6]. The lower manipulation rate could be explained by a difference in the definition of drug manipulation. Moreover, in the Netherlands, 30% of the pediatric outpatient drug administrations had to be manipulated before use [7]. Dosing accuracy and bioavailability can potentially be negatively influenced by such manipulation of the drug product [5,9].

Another portion of drug administrations in the pediatric population is through extemporaneous, or magistral, preparations. Often, these consist of oral liquids with certain disadvantages, such as stability issues, bad taste, harmful excipients and risk of dosing errors. There is a lack of insight into the extent to which pharmacy preparations are used to treat children. Most prescriptions entail marketed drugs and the percentage of overall prescriptions that need to be compounded varies from <1% to 10% with a high variability, probably due to demographic characteristics [10,11,12,13]. A survey conducted in hospitals in Japan found that 9.6% of all pediatric oral prescriptions needed to be compounded [14]. Interestingly, pharmacy preparations seem to make a comeback as drug policy shifts towards personalized medicine [15,16]. Personalized medicine asks for tailored medication which can be provided by compounding. Besides an individualized dose, other specific patient needs, such as sensory processing disorders, food allergies or dietary needs, can be taken into account when compounding personalized medicines [17]. However, compounding brings along the risk of contamination, supra- and subtherapeutic errors [18]; moreover, it demands highly trained personnel and premises which are no longer available in all pharmacies.

It is obvious that there is still an unmet medical need in the pediatric population for suitable dosages, dosage forms and formulations. The use of three-dimensional (3D) printing of personalized medicine holds the promise to aid the development of pediatric drugs [1,19,20]. Its principle is based on building an object layer-by-layer onto a printing plate from a computer model. Using computer-aided design (CAD), the model can be adjusted to meet the user’s requirements. Various 3D printing techniques can be employed to produce flexible dosage forms in terms of dosage, geometry, drug release kinetics and composition [21,22]. Extrusion-based 3D printing techniques are most often employed. Examples of extrusion-based 3D printing techniques are fused deposition modelling (FDM) [23], semi-solid extrusion (SSE) [24] and direct powder extrusion (DPE) [25]. Extrusion-based 3D printers extrude a molten or semi-solid formulation through a nozzle onto the printing plate. They can be either heated or non-heated, depending on the specific technique and formulation. Powder-solidification 3D printing techniques bind powder particles together with a binder fluid or through sintering. They include drop-on-powder (DoP) and selective laser sintering (SLS) [26]. Inkjet printing is a drop-on-demand method. Small droplets are deposited in a specific place, usually onto a substrate [21]. Finally, liquid-solidification techniques, also known as vat photopolymerization, entail amongst others stereolithography (SLA) and digital light processing (DLP) [27]. A liquid formulation is solidified in specific places to create the desired model.

It is because of the high flexibility that 3D-printed medicines are a promising asset in the treatment of the pediatric population, as was also previously mentioned in the mini-review by Preis and Öblom [19]. Indeed, these innovative developments are studied intensively both by pharmaceutical companies and universities.

There is a need for a comprehensive and timely overview of literature on 3D-printed medicine in the pediatric population. This scoping review aims to provide an overview of the developments, possibilities and limitations of the 3D printing technique for the production of pediatric drug formulations. As a secondary objective, it aims to provide a roadmap towards the integration of 3D-printed medicine in the daily treatment of pediatric patients.

## 2. Methods

For this scoping review, identification and selection of relevant literature were performed using the population/concept/context (PCC) framework [28]. The population regarded children of all ages, including the age of 18 years old. Any article not including a defined population or that is singularly aimed at the adult population was excluded. The concept aimed at medication produced with a 3D printing technique. All variations on the 3D printing terminology and different techniques were eligible for inclusion. Articles where the active pharmaceutical ingredient (API) was added after the 3D printing process were excluded, as well as articles exclusively on regenerative medicine and medical devices. Otherwise, any type of medication, in any kind of dosage form and route of administration, was eligible for inclusion. Any kind of context was deemed of interest. The overview of in- and exclusion criteria is provided in Appendix A.

The Preferred Reporting Items for Systematic reviews and Meta-Analysis extension for Scoping Reviews (PRISMA-ScR) guideline was used for the preparation of the report [29]. An experienced librarian conducted comprehensive literature searches of electronic library databases on the 8th of August 2021. Databases that were consulted, were PubMed, MEDLINE, Embase, Web of Science, Cochrane Library, Emcare, Academic Search Premier, Google Scholar, IEEExplore and ACM Digital Library. Keywords and medical subject headings (MeSH) were identified for the search strategy together with one of the reviewers. The search strategy was initially set up for PubMed and adapted for the other databases. The full search string is provided in Appendix B. Any type of full-text article was eligible for inclusion. No restrictions were made regarding publication year and study design. Only English written publications were included. Reference lists of eligible articles were manually screened for relevant cross-references. Article selection was checked by a second reviewer. Any discrepancies were resolved by consensus, if necessary, with a third reviewer.

The data extraction followed the PCC framework. This means that included articles were screened on population characteristics, concept characteristics and context characteristics. Of particular interest were population age, treatment indication, drug treatment, qualitative and quantitative composition of formulations, quality control requirements and outcomes, printing technique, printer settings, safety considerations, patient acceptance and regulatory considerations.

## 3. Results

### 3.1. Literature Selection

A total of 1613 records were identified in the initial search on the 8 August 2021. Of these records, 738 duplicates were removed. Of the remaining records, the title and abstract were screened for relevance. A further 716 records were excluded, as they did not meet the inclusion criteria. Another 23 reports could not be retrieved, all of which were reviews on 3D printing of medicine not specific for the pediatric population. The full-text of the remaining reports were screened for eligibility. Two reports were excluded, as they pertained only an abstract and 92 were excluded based on the inclusion criteria. A total of 42 reports were included, of which 33 were original research articles, 4 reviews, 2 news articles, 2 policy papers and 1 viewpoint article. During the process, seven records were discussed with the second reviewer. Consensus was reached to exclude four of these records, one record was scored as a different article type and two were included. The flow of the inclusion of the reports is represented in Figure 1.

### 3.2. Pediatric-Centered Formulation Design

When developing appropriate formulations for pediatric patients, a couple of key attributes should be considered. These key attributes can be defined in a pediatric quality target product profile (pQTPP) and consist of the route of administration, patient age range, target release profile, dosage form, dose and dose flexibility, patient acceptability, dose preparation and manipulations, dose administration device, safety of excipients, child-resistant packaging, stability and storage conditions, ease of manufacturing, and patient access [1]. The flexibility that 3D printing technology offers is of special interest for technical key attributes. These are dose accuracy and dose flexibility, target release profile, dosage form, patient acceptability and dose administration. An overview of the investigated key attributes in the identified research articles in which a formulation is studied is provided in Table 1.

#### 3.2.1. Dosing and Drug Delivery Strategies

In all identified studies pediatric drug formulations were designed that were intended for the oral route of administration. Dose accuracy, dose flexibility, target release profiles and drug delivery optimization were investigated in various studies. An overview of the studied key attributes is provided in Table 1.

Dose accuracy was defined with content uniformity [30,31,32,33,39,52,54] or drug content assay [34,35,36,40,41,42,43,44,45,55,56,57,58]. Alternatively, one study determined the uniformity of dosage units using the mass variation method of the European Pharmacopoeia (Ph. Eur.) instead of using the content uniformity method [46]. These methods are employed to assess the accuracy and precision of the production method, where the accuracy refers to the ability to produce a target dose, while precision refers to the ability to repeatedly produce a dose with a small variability. It should be noted that, while assay data are sufficient to demonstrate precision of the production process, it does not show the ability of the process to accurately produce a target dose. This was also noted by Öblom et al., who found an adequate dose precision for most batches, but not always an adequate dose accuracy [33]. Two studies found that their model drug had a lower drug content than expected. In one study, the drug content was thought to be lower due to crystallization of the model drug and abrasion during the printing process [46]. In the other study it was thought that the drug degraded during the production process [47]. Overall, the dosing precision was adequate, irrespective of the printing method used, though ink-based printing methods could be of specific interest for low-dose drugs [52,60].

Dose flexibility was studied in several ways. Most often, the dimensions of the computer model were used to correlate to the dosage strength. The theoretical volume of the computer model correlated with the dosage strength in several studies [30,32,36,52,53]. This was feasible for both high- and low-dose drugs [52]. While the definition of low- and high-dose is debatable, this study specifically looked at the absolute doses of 2.0–12.0 mg of metoprolol tartrate and 80–240 mg of theophylline per printed tablet, with respective API concentrations of 10% *w*/*w* and 35% *w*/*w*. Similarly, the printed mass could be correlated to the dosage strength [33,39,53]. Adjusting the amount of printed filament could also be used to accurately adjust the dosage strength. For instance, in one study, paracetamol and ibuprofen dose could be controlled by reducing or multiplying the printing path in a range of 25–300% [34]. Another study successfully investigated the correlation between the printed filament length and the printed filament mass [48]. The number of printed layers could also be correlated to the dose of the printed tablets [31]. Indeed, adjusting tablet model dimensions to control the drug dose has been extensively studied. Furthermore, it has already been used in clinical practice in a study treating children with maple syrup urine disorder (MSUD) [40].

Dose flexibility has also been achieved by altering the drug concentration in the tablet matrix [39,49], though a higher drug concentration could reduce the printability of the tablet matrix [49]. For inkjet printing and drop-on-solid 3D printing, the concentration of the drug in the printing ink can be altered to adjust to printed dose [52,55,56]. Furthermore, for inkjet printing, the printing size [33,55,59], the printing resolution [54] and the number of printed ink layers [55,58,59] determined the drug loading on the edible film. One study also determined the feasibility of using the number of printed ink layers for drop-on-powder 3D printing [52]. Several studies found that using the number of drug-containing layers as a dose flexibility method is less accurate [52,55,58]. This can be explained by the difference between the theoretical number of layers, which could be a decimal number, and the actual number of printed layers, which has to be an integer [61].

Specific drug release profiles can be produced using 3D printing techniques. Several studies show that an increase in tablet dimensions lead to a reduced drug release rate [46,50]. While the infill percentage, the ratio of tablet matrix to air within the drug product, causes a smaller effect on the release rate [50], this can also be used to control the dissolution and, therefore, the release kinetics of the drug product [42,47,50]. Furthermore, the drug release rate can be altered by adjusting the formulation [52]. In one study, corn starch was added to the formulation and found that it acted as a gelation agent. It, therefore, reduced the release rate [44]. While most sustained release formulations were produced using a heated extrusion-based 3D printer [42,44,46,50], one group used a non-heated extrusion-based 3D printer [34] and another group used a DoP 3D printer [52].

Finally, heated extrusion-based 3D printers can enhance solubility properties of poorly soluble drugs. In one study, hot-melt extrusion (HME) followed by DPE 3D printing was used to enhance the drug release of 100 and 150 mg praziquantel printed tablets after 2 h by more than four-fold as compared to pure praziquantel [49]. Similarly, another study improved the dissolution of rufinamide. When using a dose of 1600 mg, they found an increased dissolved amount after 2 h compared to commercially available tablets [43]. Both studies held the amorphous state of the drug in the formulation accountable for the enhanced solubility.

#### 3.2.2. Acceptable Tablet Size

As marketed drugs are often too large for children to swallow, a good alternative solution is small sized tablets. The maximum acceptable diameter of minitablets is dependent on the age of the patient, though the proposed maximum is 5 mm [62]. Tablets produced with a 3D printer that were deemed by the authors to be an acceptable size for their intended target age group ranged in diameter from 1.5–10 mm [46,47,50]. Smaller tablets tended to exhibit a less reproducible printing shape, yet they would still have an adequate dose accuracy [46]. In another study, tablets with larger dimensions showed a reduced dissolution rate. The small capsule-shaped tablets possessed predefined dimension ratios, with lengths of 5.0, 7.5 or 10 mm [50]. While their smallest tablet is within the proposed maximum size, it is arguable whether the other tablet sizes are acceptable for use in children. Similarly, a third study produced tablets with dimensions of 9 × 5 × 4 mm. The authors argued that the tablets were intended for use in children no younger than the age of 6 years old [47].

#### 3.2.3. Palatable Oral Dosage Forms

With 3D printers, oral dosage forms can be made more appealing for children by producing tablets with eye-catching appearances and favorable palatability. Some studies produced chewable dosage forms, while others focused on taste-masking. In a clinical study, chewable tablets have already been used. Round tablets with different sizes, colorants and flavorings were produced to match the patient’s needed dose and preferred color and flavor [40]. However, chewables could also take candy-like forms. Medicinal gummies or jellies are an example of chewable dosage forms [34,37,44]. They were produced in different shapes, even that of a LEGO^®^-like brick [34], and various food colorings and sweeteners could be added to the formulation. Formulations shaped in different cartoon- and candylike figures were also made [38,41,51], of which one was a chocolate-based dosage form [41].

Chewable dosage forms were most often produced with SSE or food 3D printers. The excipients used for these techniques are possibly more suitable for chewable dosage forms. Only one study used an FDM 3D printer to fabricate Starmix^®^-shaped tablets [51]. The chewable dosage forms in the studies were relatively large when compared to conventional tablets, with a minimal measured diameter of a round tablet of 8.2 mm [40] and the largest length of a figure-shaped tablet being 84.1 mm [41]. While size of the dosage form is less critical for the acceptability than it would be for tablets that must be swallowed whole, it should still be taken into consideration. The FDA recommends the same maximum size for all dosage forms, meaning the largest dimension should not exceed 22 mm [63].

Palatability of chewable dosage forms was determined through different methods. The chocolate-based formulations were tested for their mouthfeel properties through textural analysis. Furthermore, the drug release showed to be 22.86% for paracetamol and 36.41% for ibuprofen in simulated saliva fluid within two minutes [41]. While this could possibly lead to an enhanced uptake of the drugs, a higher release in saliva can also effectuate the bitter flavor of the drugs. Two other studies employed healthy volunteers for taste evaluation. Indomethacin 3D-printed tablets were held in the mouth then spat out and compared to the taste of pure indomethacin. Pure indomethacin possessed a moderate bitterness, while the 3D-printed tablets were scored to possess threshold bitterness [51]. Wang et al. let students taste formulations with various flavoring agents to determine the most palatable formulation [38]. Finally, in a clinical study, the acceptability of chewable tablets was evaluated by patient and parent reported outcomes, though the population was too small to draw conclusions on the palatability when comparing the different flavors with each other and compared to compounded capsules [40].

Another method of improving the palatability of tablets is through taste-masking. Delaying the drug release from the tablet matrix until after the tablet has passed the mouth, can prevent bitter tasting drugs from getting in contact with the taste buds on the patient’s tongue. One way of delaying the initial drug release is by producing a tablet in which the drug is in an amorphous state or is molecularly dispersed. Two studies have conducted research on taste-masking by enveloping a bitter tasting API in an amorphous polymer. They first produced filaments with HME. One study then used these filaments, which contained the bitter model drug caffeine, directly in an FDM 3D printer to produce donut-shaped tablets [42]. In the other study, the praziquantel containing filaments were further processed prior to use in a DPE 3D printer [49]. The effectiveness of the taste-masking was tested by performing adjusted in vitro dissolution testing using simulated saliva fluid. The concentration after 10 min of testing should not exceed the bitterness threshold concentration for successful taste-masking. Both studies concluded their formulations were sufficiently taste-masking [42,49]. Furthermore, when comparing the dissolution of the 3D-printed tablets in artificial saliva to tablets produced with direct compression, the directly compressed tablets fully released the drug in 50 s. This indicated no taste-masking of the directly compressed tablets [42].

#### 3.2.4. Orodispersible Dosage Forms

Ease of administration can be improved by using orodispersible tablets or films. Since 2015, 3D-printed tablets containing levetiracetam in doses up to 1000 mg have been approved by the FDA for use in patients from the age of 4 years old [64,65]. Spritam^®^ tablets are orodispersible tablets intended for oral suspension. It is produced using a DoP 3D printer. Due to the loosely bound powder particles with a binder fluid, the tablets are orodispersible. In resonance of the approval of Spritam^®^, one research team developed a DoP 3D printer which utilizes multiple printing heads. As a result, they managed to produce various colorful cartoon-like orodispersible levetiracetam tablets. The quality controls for hardness, friability, dispersion uniformity and release characteristics were not inferior to the Spritam^®^ tablets [53]. Another study also used levetiracetam as a model drug, but used an extrusion-based 3D printer to produce orodispersible tablets. By using a highly water-soluble polymer as the tablet matrix, the tablets, irrespective of their size, were able to disintegrate within 3 min and could, therefore, be classified as orodispersible as per definition of the Ph. Eur. They found that the disintegration time is dependent on the number of printed layers in the tablet [31]. Another group produced orodispersible tablets containing hydrochlorothiazide with a SSE 3D printer. Their tablets disintegrated within 3 min by reducing the infill percentage of the tablets to 70% [35].

Orodispersible films were typically produced using inkjet printers [33,54,55,56,57,58,59], or with extrusion 3D printers [32,33,36,45]. The drug containing ink is printed onto a carrier film layer. A multitude of drugs have been printed onto orodispersible films. Interestingly, while orodispersible tablets typically contain a high drug load, orodispersible films, especially those produced with inkjet printers, are more suitable for low drug loads. This reflects the versatility and limitations of extrusion-based 3D printers and inkjet printers.

Inks, or binding fluid in DoP 3D printing, were typically used for the production of orodispersible dosage forms. Volatile solvents were used in these inks to obtain solid dosage forms after a drying step. Solvents found to be used in the production of orodispersible dosage forms were isopropanol [53], methanol [56,57], acetone [57], ethanol [33,36,57,59] and water without additional solvents [31,35,59]. One study used edible ink as a base ink, which contained propylene glycol [58]. Another study also used propylene glycol as a viscosity modifier and plasticizer in their ink formulation. However, they argued that the residual content should be determined and propylene glycol should preferably be replaced with pediatric suitable excipient [33]. Residue testing was only performed by one study. They tested for the residuals of acetone, which was used as solvent for the carrier film, and methanol, which was used as solvent for the drug containing ink. No residual methanol could be detected with a method validated at 0.3 ppm, which is well below the limit of 3000 ppm stated in Ph. Eur. 5.4. The casted hydrochlorothiazide films contained 470 ppm acetone, which possesses a compendial limit of 5000 ppm, though this limit might be unsuitable for children. [57]. Though not specific to printing drug products, it does reflect the necessity of residual solvent testing of orodispersible films.

### 3.3. Clinical Implications

#### 3.3.1. Clinical Application

Two studies that applied 3D-printed medicine in the pediatric population were identified. One group treated inpatients aged <1 day up until 9 months with spironolactone 2 mg produced with an extrusion-based 3D printer [30]. They investigated the use of 3D-printed tablets instead of the standard of care, which were split spironolactone tablets. The patients were treated for a duration of 1 day up to 5 days. The 3D-printed tablets were dissolved in a bit of water prior to being taken by the patients. Any clinical outcome measures were not mentioned in the article.

The other group produced personalized isoleucine chewable tablets with an SSE 3D printer for four pediatric patients aged 3–16 years diagnosed with MSUD [40]. The needed dose of isoleucine was determined based on the isoleucine concentration levels obtained from dried blood spot samples. The isoleucine dose in the tablets was adjusted by altering the tablet size. The tablets contained 50 mg, 100 mg, 150 mg or 200 mg, depending on the required dose for the specific patient. Patients were treated for 3 months with the 3D-printed tablets. Adequate isoleucine concentration levels could be maintained by using 3D-printed chewable tablets. Differences in clinical outcome between the compounded capsules and the 3D-printed tablets could not be established due to the small population size.

3D-printed levetiracetam tablets, Spritam^®^, have been approved by the FDA since 2015 [64,65]. These tablets are, therefore, used in clinical practice. While no Spritam^®^-specific studies have been published with clinical data in the pediatric population, a bioequivalence study in healthy adults has been published [66]. They found that the 3D-printed tablet was bioequivalent to the reference drug under fasted conditions. So far, Spritam^®^ is the only commercially available 3D-printed preparation.

#### 3.3.2. Patient Acceptability

Both aforementioned clinical application studies found adequate acceptability of the 3D-printed tablets by patients [30,40]. Goyanes et al. evaluated the acceptability of different flavor-color combinations using qualitative research methods [40]. Sample formulations were scored by patients on a five-point facial hedonic scale, with higher scores representing higher acceptability. Parents scored the facial expression on a scale of 1–3. All tablets were well accepted and the orange flavored and colored tablets received the highest score. However, it was not possible to determine which flavor and color were most and least favored, due to the small patient population. Furthermore, it could not be established whether the 3D-printed tablets were better accepted than the compounded capsules, as most patients did not swallow the capsules whole.

In another qualitative study the acceptability of different kinds of 3D-printed tablets was described through a visual preferences survey [67]. Different placebo tablets were printed using either a DLP, SLS, SSE or FDM 3D printer. Primary school children aged 4–11 years old were asked to record which tablet they liked best and which one they thought was the worst. Then they were asked why they preferred or disliked the tablets. The DLP tablets were chosen most often as best liked, while the FDM tablets were least chosen as best liked. SSE tablets were scored the worst liked most often, and DLP tablets were scored the worst liked the least often. Interestingly, when the children knew the SSE tablets were chewable, 79% of the participants would chose the SSE tablets as best liked and, therefore, it became the most favored tablet. Themes that were thought to be of importance to children were appearance, perceived taste, texture and familiarity.

#### 3.3.3. 3D versus Conventional Manufacturing

Eight studies have been identified that explored the quality differences between 3D-printed formulations and dosage forms manufactured in a traditional way, so either marketed products or compounded dosage forms.

The clinically applied 3D-printed tablets were compared to compounded capsules [40] and subdivided tablets [30]. 3D-printed tablets containing 2 and 4 mg spironolactone and 5 mg hydrochlorothiazide were visually an improvement to subdivided tablets split by pharmacists. The split tablets exhibited an irregular size and rough surfaces, while the shape of 3D-printed tablets was uniform and smooth. Moreover, the content uniformity was improved. 3D-printed spironolactone 2 mg and 4 mg, and 3D-printed hydrochlorothiazide 5 mg contained a drug content of 100.89 ± 2.09%, 98.54 ± 1.96% and 98.69 ± 1.80%, respectively. In comparison, the subdivided tablets contained a drug content of 133.46 ± 20.45%, 107.50 ± 10.90% and 81.56 ± 13.91% for spironolactone 2 mg and 4 mg, and hydrochlorothiazide 5 mg, respectively. The corresponding calculated acceptance value of the content uniformity was well within the requirement limits for all 3D-printed tablets, while they were out of specifications for subdivided tablets. Commercially available tablets and 3D-printed tablets had comparable dissolution profiles [30]. No difference in acceptability and clinical outcomes could be established for 3D-printed tablets compared to compounded capsules [40].

One group investigated the impact of three different dose dividing methods on the dose accuracy and dissolution rates of theophylline 80 mg and metoprolol tartrate 5 mg as compared to 3D-printed drug formulations produced with a DoP 3D printer [52]. They concluded that 3D printing of either API resulted in the most accurate drug dose, followed by the tablet liquefaction. Tablet grinding and packaging the resulting powder ranked third, while tablet splitting was the least accurate. They further concluded that 3D-printed tablets exhibited dissolution profiles comparable to commercially available tablets. For the theophylline tablets, the dissolution rate was increased for the powder and liquefaction dose dividing methods, while the commercial tablets were marketed with a sustained release profile.

Two other research teams also compared the dissolution profiles of their 3D-printed tablets to commercial tablets. One study aimed at developing 3D-printed tablets with a comparable dissolution profile, at which they succeeded with the lowest infill percentage of 65% [47]. In the other study a formulation was developed with improved dissolution kinetics compared to the commercially available tablet [43]. At low doses of rufinamide, a water-insoluble drug, the commercial tablet showed faster dissolution rates. When using a higher dose of the drug, the 3D-printed tablet showed an improved dissolution profile, as more drug could be dissolved in the dissolution medium overall.

Orodispersible films were printed with inkjet printers, also referred to as 2D printers, and extrusion-based 3D printers, usually SSE printers. One study compared orodispersible films made with both techniques and they compared them with compounded oral powders [33]. They found that 3D-printed orodispersible films exhibited a faster disintegration time than 2D printed films. This was attributed to the thickness of the film and the increased surface area of the 3D-printed films due to extrusion structure of the film. The oral powder showed the fastest dissolution rate as compared to 2D and 3D-printed films. Furthermore, both 2D and 3D-printed films exhibited an improved dose precision compared to oral powders. 3D-printed films complied to the requirements for content uniformity, and therefore dose accuracy, most often.

#### 3.3.4. Implementation of 3D-Printed Medicine

One study was found in which pediatric healthcare professionals held focus groups discussions about (1) 3D printing as a manufacturing technology for drug products and (2) the need for personalized medication [68]. Four main themes were identified. These were the benefits of 3D-printed drug products, concerns regarding 3D printing, prerequisites for adoption at hospitals, and suggestions for printed medicines. Recognized benefits were on-demand personalized dosing, personalized dosage forms, ease of drug administration, medication safety, polypills and possible cost savings. Concerns regarded medication safety, drug administration, on-demand production and delivery, and costs. Prerequisites were identified to address medication safety, drug administration and on-demand production and delivery. To ensure medication safety, 3D-printed medication needs to have adequate product quality and stability, product identifications need to be possible and drug interactions need to be accounted for by pharmacists. With regards to drug administration, 3D-printed products need to have a suitable size for children, should preferably be dissolvable or dispersible and administration through an enteral feeding tube should be possible. Furthermore, logistically, there must be a short response time for production and delivery to meet the need for treatment that must start quickly. Lastly, the healthcare professionals provided suggestions for 3D-printed products that they thought are needed.

#### 3.3.5. Eligible Active Pharmaceutical Ingredients

The healthcare professionals from the abovementioned study suggested multiple API’s for which there currently is a lack of oral products, a need for oral personalized doses, a need for orodispersible drug products, a need for combination products, or a need for better treatment options [68]. Candidate API’s for 3D-printed medicine for the pediatric population were also suggested in a viewpoint article [69]. The authors suggest focusing the development of 3D-printed medication firstly to API’s that lack a suitable oral formulation. Secondary candidates are API’s that are now formulated in oral liquids with poor acceptability and large tablets. Finally, API’s should be considered for which the current formulation can be improved.

One group researched the suitability of API’s for inkjet printing on orodispersible films [60]. From their database, they extracted a total of 612 different API’s that were prescribed to children aged 0–5 years in a timespan of five years. Exclusion was based on incomplete ATC code, no legal status, duplicates, drug load >50 mg, API’s for other administration routes than oral, off-label use and if an oral liquid was available on the market. As such, 34 API’s were selected as being suitable. It should be noted that this study specifically focused on inkjet printing, and so API’s will have been excluded, which would be of interest for other printing techniques.

Various studies substantiated the medical need for choosing a specific API. Most often the current need for manipulation or preparation of oral drugs was mentioned [30,32,33,40,48,49,50,52,54]. In extension, the current lack of personalized oral dosages were also often mentioned [31,36,55,56,57,59]. Furthermore, poor patient acceptability of the current treatment options [38,39], improving currently available formulations [43,44], drug combinations [34] and the lack of oral drug products [45] were mentioned as reasons for choosing the model drug.

#### 3.3.6. Target Population and Disease

The identified model drugs were often indicated for the treatment of cardiovascular events and diseases [30,33,35,36,39,46,48,52,54,57,59]. Drugs used for the treatment of epilepsy and related diseases [31,37,43,53,56], and pain reduction and peri-operative treatment drugs [34,41,45,51] were also often used as model drugs in the studies. Other diseases for which a printed formulation was developed, were asthma [55], nutrient deficiency [58], MSUD [40], muscle relaxation [50], peptic ulcer disease [44], spleen deficiency and stomach distension [38], allergic rhinitis [32], schistosomiasis [49], apnea [46,52], and malaria [47]. Not all authors explicitly stated the intended use of the model drug, but derivations can be made as the drugs have specific indications.

The intended age group for whom the formulations were developed were not always specified. All age groups, as defined by ICH guideline E11 [70], were represented in the articles. Formulations for newborns (age 0–27 days) were orodispersible or readily dissolvable dosage forms [30,31,54]. Interestingly, only one study produced tablets of an acceptable size for neonates, though they did not specify their target age group [46,71]. Most formulations intended for infants and toddlers (age 28 days–2 years) were also orodispersible or readily dissolvable [30,31,32,39,55,56]. For children (age 2–11 years) chewable dosage forms and swallowable dosage forms were developed [31,32,39,40,47,51,56]. Finally, for adolescents (age 12–16/18 years) orodispersible films, a chewable tablet and a tablet to be swallowed whole were developed [32,40,47,56]. Interestingly, only three formulations were developed that could be applied in three of the four age groups [31,32,56]. All of these were orodispersible dosage forms.

## 4. Discussion

A lot of research has already been performed on 3D-printed pediatric drug formulations. Furthermore, looking at the key attributes for pediatric-centered product development, multiple attributes have already been investigated in these studies. However, for each attribute improvements can still be made.

All dosage forms in the found literature were intended for the oral route of administration. This is the most attractive route of administration for all patients, the pediatric population not exempted. However, should the oral route not be available for drug intake, different routes should be possible. 3D-printed drug products with different routes of administration have also been previously investigated. Examples are drug-eluting structures [72] and suppositories [73]. To facilitate adequate drug delivery, with any route of administration, various methods can be employed to regulate drug release. Not only can the preparation matrix be chosen for a specific release profile, also a shell or coating could be printed around the drug containing core. This enables complex drug release systems with sustained, delayed or pulsatile drug release [74].

As described in the results, oral dosage forms can be printed with precise, accurate and flexible doses. This has multiple advantages for pediatric-centered formulation development. Personalized and flexible dosing is the most important advantage of 3D-printed pharmaceuticals. Manipulation and preparation of dosage forms can become largely unnecessary as 3D-printed dosage forms can fill the gap. 3D-printed dosage forms have a more accurate dose than manipulated tablets [30,33,52]. Though the found literature has provided data on the suitability of 3D-printed tablets as compared to manipulated tablets, it has yet to be established for other oral administration dosage forms, such as compounded oral liquids. The use of this dosage form is limited by the often-unknown stability of the formulation, unpleasant taste, use of potentially harmful solvents and dosing errors by the user or caregiver [3,75]. The stability of 3D-printed formulations might be better than the stability of oral liquids, as fewer physical, chemical and microbiological risks can be expected due to lack of water and restricted movability of molecules. However, not much is known on the stability of 3D-printed tablets. Though stability and residual solvent data are still largely missing for 3D-printed tablets, it has already been proven that they can have an acceptable palatability and dosing precision and accuracy.

Furthermore, 3D printers offer an advantage in the number of manufacturing steps compared to conventional tablet manufacturing. No milling, granulation or compression steps are needed. This can make 3D printing more suitable for small batch tablet production than conventional tablet manufacturing. However, manufacturing challenges of 3D-printed drug products concern, amongst others, drug particle size and nozzle diameter control, mixing and drying steps, which can reduce the production efficiency. Whether 3D printing is more time efficient, including in comparison to conventional compounding methods, is furthermore dependent on the formulation starting point. Preparing the whole formulation can be time consuming. Another starting situation could be a basic solution or filament in which the API is integrated in the desired concentration at the production facility. Such a general basic solution or filament should prior be tested for compatibility with a specific API, comparable to what is currently is done for an API in general suspension liquids. Finally, a fully standardized drug-containing cartridge or filament can be used as a starting point. In this last case, the production efficiency is improved when compared to conventional compounding methods [33], enabling true on-demand manufacturing. However, few research articles have been found on the manufacturing and storage stability of these intermediate products [48,76]. Challenges remain with the storage expiration date. Long-term stability of intermediate products or 3D-printed dosage forms is not yet known. The stability of intermediate products should be sufficient. Pharmacists should be able to store the intermediate products for an adequate amount of time before dosage form production to be cost-effective [77].

Cost-effectiveness is imperative to successfully implement 3D-printed drug products, as it enhances patient accessibility. Furthermore, regulatory guidelines and non-destructive quality control requirements are yet to be established for 3D-printed drug products, but are needed to ensure user-safety, legal protection and waste-reduction [39,78,79]. The regulatory viewpoint might depend on the starting point of the 3D printing process. If the formulation is fully compounded at the production facility, it might be regarded as compound manufacturing. This requires a specialized production facility, such as a compounding pharmacy or at the point of care in a hospital pharmacy with adequate equipment. However, if an intermediate product is produced by the pharmaceutical industry, it can be considered to officially market the intermediate product. Community pharmacies can then also more easily integrate a 3D printer in their daily practice. A change in the regulatory landscape is needed to be able to do so. Regardless of the formulation starting point, regulatory viewpoint and used type of 3D printer, the production process must be validated in order to ensure patient safety.

While abovementioned reasonings might not be specific to the pediatric population, there are ethical issues for pediatric-centered 3D-printed drug formulations that should be considered by the pharmaceutical product developers and be reflected in guidelines of regulatory authorities. An important factor in treatment compliance by children is the patient acceptability. Palatability is regarded as one of the most important factors in patient acceptability. It comprises various organoleptic properties as appearance, smell, taste, after-taste and mouth-feel [80]. Improving organoleptic properties will inherently lead to an increased willingness of pediatric patients in taking their medicine, but also in being able to mistake these drug products as candy [68,81,82]. In this light, any formulation that can be perceived by children as candy should be carefully considered before prescribing such a drug product. As is clear from the results of this review, quite some research has already been performed on the ability of producing candy-like 3D-printed drug formulations for pediatric use. These products should be treated with caution as to prevent accidental intake of drugs by children.

Nevertheless, suitable palatability is needed for the pediatric population and is extensively investigated in the identified literature. What is considered suitable palatability can differ between pediatric age groups, as with increasing age the desired dosage form and perceived taste changes [83,84]. Moreover, palatability, and especially bitter taste, is related to patient factors such as the genetic constitution of the individual patient [85]. The younger the patient, the smaller the dosage form should be, whether it is a solid oral dosage form or a liquid [86]. Interestingly, minitablets as small as 2 mm were accepted by neonates [71]. Furthermore, small tablets are better accepted by infants, toddlers and preschool children than oral liquids are [87,88]. However, there is little known about the possibility of 3D-printed minitablets. The accuracy of 3D-printed minitablets should be established further. Apart from the minitablets described in the results, minitablets have also been produced using an SLS 3D printer [89]. However, the applicability of this technique is limited to API’s and excipients that are not photo- and thermosensitive [26]. Development of 3D-printed minitablets can be improved by focusing on higher resolution of extrusion-based techniques and increasing the drug load in the tablet matrix. Though increasing the drug load reduces the printability of the tablet matrix. High drug loads of up to 80% *w*/*w* have been reported for extrusion-based techniques, where are, therefore, possible [35,90,91,92] and should be further explored for pediatric formulations.

For orodispersible dosage forms, which were frequently identified in this review, there is little evidence for their acceptability in neonates, infants and toddlers. For school-aged children and adolescents orodispersible and chewable dosage forms might be a suitable alternative for conventional tablets [83,93], though this should be further substantiated. In theory, however, orodispersible and chewable dosage forms are very attractive for children. Even though the dosage forms are orodispersible or chewable, the size should still be appropriate for use in children. Healthcare professionals have raised their concerns regarding the maximum size of the dosage forms [68]. The FDA has stated that the same guideline for maximum size of tablets applies to all oral solid dosage forms [63]. Therefore, efforts should be made to further reduce the size of 3D-printed orodispersible and chewable dosage forms.

An advantage of 3D-printed pharmaceuticals is that they may be taste-masking, therefore improving the palatability of the dosage forms [42,49]. The bitter taste can be masked by molecularly dissolving the API in the tablet matrix. Another advantage that comes with solid dispersions is the enhanced bioavailability of poorly water-soluble API’s [94]. This could lead to better efficacy, a lower needed dose and fewer side effects. However, improving the bioavailability of established API’s also poses a risk of overdosing. Prescribers and pharmacists should be aware of the possible implications of 3D printing on the bioavailability of API’s. Furthermore, investigations must be performed to improve the knowledge of the safety and efficacy of molecularly dispersed API’s in 3D-printed dosage forms.

When considering suitable dosage forms for pediatrics, the safety of excipients should also be taken into consideration. As previously described in this review, orodispersible dosage forms often need solvents for their production. These excipients can be toxic to children [95]. They should therefore preferably be avoided in pediatric formulations. If they need to be used, it should, therefore, be ensured that these excipients are evaporated adequately from the dosage form prior to administration. Only one article was identified in which the residual contents of the excipients were determined [57]. Limited availability of suitable excipients also inhibits the use of liquid-solidifying techniques [27] and SLS [26]. Selecting suitable excipients for pediatric formulations can be challenging, as safety data for pediatric patients is not always available. However, key unsuitable excipients are clearly defined. Furthermore, the European Paediatric Formulation Initiative (EuPFI) provides the Safety and Toxicity of Excipients For Paediatrics (STEP) database [96], which holds available data on excipients that might be used in pediatric formulations.

Finally, different printing techniques were identified in this review. These techniques each had their own advantages in the production of pediatric medicine. Inkjet printing is specifically of interest for low-dose drugs, as it is a high precision technique, but is also limited in the amount of drug that can be printed onto a substrate. DoP 3D printers, on the other hand, are more suitable for the production of high-dose drugs. These printers, however, require a solvent and the mechanical properties of the resulting dosage forms can still be improved. A lot of experience has been gained with pharmaceuticals produced with an FDM 3D printer [23,97,98,99]. This technique is considered a low cost and easy to use technique. However, FDM usually has a high operating temperature, so the processability of thermolabile API’s is limited with this technique, though not impossible with low temperature FDM [100,101]. SSE 3D printers operate at low temperature, if even at all heated, and are, therefore, suitable for a wide range of API’s and excipients [102]. It is of particular interest for the clinical setting. Different printing techniques can, therefore, be used for different purposes. There is not one technique specifically suitable for pediatric formulations, though suitable excipients, target dose and physicochemical properties of the API can limit the suitability of a 3D printing technique.

From the present review, it can already be concluded that 3D-printed drug products can and should be considered as a serious alternative to compounded drug products. It is the responsibility of the pharmacist to assess the most suitable manufacturing method should a marketed drug product not suffice for an individual patient. 3D-printed drug products have proven themselves to be superior in terms of drug dosing and release profile compared to conventional compounding methods. To strengthen the foothold of 3D-printed drug products, a good manufacturing practice (GMP) ready 3D printer should be readily available in pharmaceutical production facilities. Furthermore, compatibility and stability data of the API in the intended tablet matrix should be known before use of 3D-printed drug products. These conditions enable the use of 3D-printed drug products as pharmaceutical preparations. Further research should focus on expanding clinical knowledge, providing regulatory guidance and expansion of the product range, including dosage form possibilities, and enabling the use of diverse GMP ready 3D printing techniques for the production of various drug products. These conditions enable the use of 3D-printed drug products for a larger patient population, thus, also the ones for whom current treatment is moderately suitable. Firstly, further clinical knowledge is needed to convince prescribers and patients of the advantage 3D-printed products offer and ensure the safety of these 3D-printed drug products. Secondly, regulatory guidelines are necessary to determine the legislative position of 3D-printed drug products, therefore providing guidance and safety for pharmacists and production specialists, and patients. Thirdly, while some API’s have been studied in 3D-printed drug products, expansion of this knowledge is warranted. The API should be selected upon necessity, so an API for which a suitable drug product is not presently available is preferred. Finally, it should be further investigated which API is most suitable for which 3D printing technique. This will enable the most effective use of 3D printing techniques for the production of drug products.

While these conditions are not exclusively applicable to the pediatric population, it is of particular necessity for children. Drug product development is often directed at adults, leaving the pediatric population to the grace of the results of adult clinical trials. This leads to the use of manipulated marketed tablets intended for the adult population, or the pharmaceutical preparation of bad tasting oral liquids. A clear medical need exists in the pediatric healthcare, for which 3D-printed drug products offer a promising option.

## Figures and Tables

**Figure 1 pharmaceutics-14-00416-f001:**
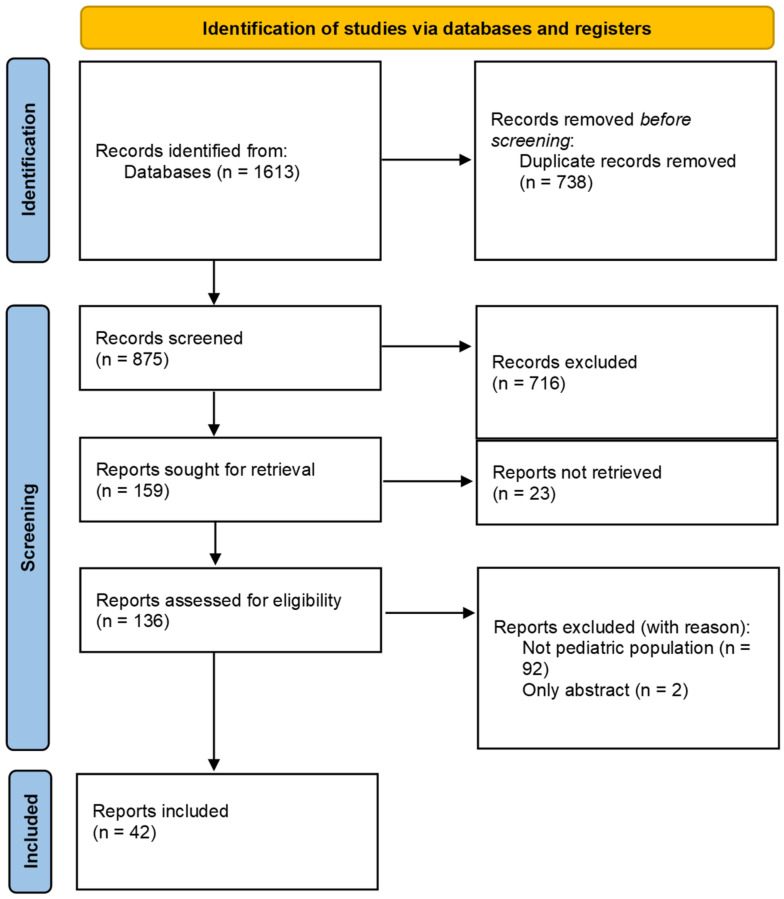
Study selection flow chart.

**Table 1 pharmaceutics-14-00416-t001:** Overview of the investigated key attributes of interest for 3D-printed oral dosage forms. Unknown or unpublished data are indicated by “-“.

Printing Technique	Dosage Form	Active Pharmaceutical Ingredient	Assessment of Dose Accuracy	Method of Dose Flexibility	Target Release Profile	Patient Acceptability	Dose Administration	Ref.
Extrusion-based; non-heated	Tablet	Spironolactone	Content uniformity	Volume	Similar to commercial tablet: immediate release	Accepted by inpatients, family members, doctors, nurses and pharmacists	Dissolved in water prior to administration	[30]
		Hydrochlorothiazide						
	Orodispersible tablet	Levetiracetam	Content uniformity	Number of layers	Immediate release	Tablet size not suitable for younger children	Orodispersible	[31]
	Orodispersible film	Levocetirizine	Content uniformity	Volume	Immediate release	-	Orodispersible	[32]
	Orodispersible film	Warfarin	Content uniformity	Volume	Immediate release	Acceptable size for children	Orodispersible	[33]
							Administration through feeding tube	
	Chewable dosage form	Paracetamol	Drug content assay	Filament length	Extended release	Acceptable palatability	Chewable	[34]
		Ibuprofen						
	Orodispersible tablet	Hydrochlorothiazide	Drug content assay	-	Immediate release	-	Orodispersible	[35]
	Orodispersible film	Warfarin	Drug content assay	Volume	Immediate release	-	Orodispersible	[36]
	Chewable dosage form	Lamotrigine	-	-	Immediate release	Various colors and shapes	Chewable	[37]
						Acceptable palatability		
	Tablet	Jiuxiang Jianpi Yangwei (JJY)	-	-	-	Cartoon shapes	-	[38]
						Acceptable palatability		
Extrusion-based; heated	Tablet	Furosemide	Content uniformity	Drug concentration	Immediate release	Size suitable for children	Swallowable	[39]
		Sildenafil						
	Chewable tablet	Isoleucine	Drug content assay	Volume	Immediate release	Acceptable palatability;	Chewable	[40]
						Color and flavor patients choice		
	Chewable dosage form	Paracetamol	Drug content assay	-	Paracetamol: immediate release	Cartoon shapes	Chewable	[41]
						Acceptable palatability		
		Ibuprofen			Ibuprofen: pH dependent			
						Pleasant texture		
	Tablet	Caffeine	Drug content assay	-	Modified release	Taste masking	-	[42]
					Dependent on infill percentage			
	Tablet	Rufinamide	Drug content assay	Multiple tablets	Enhanced drug release	-	-	[43]
	Chewable dosage form	Ranitidine	Drug content assay	-	Immediate and modified release	Candy shapes	Chewable	[44]
					Dependent on formulation	Acceptable palatability		
	Orodispersible film	Diclofenac	Drug content assay	Surface area	Immediate release	Taste masking	Orodispersible	[45]
	Tablet	Caffeine	Mass variation	Volume	Dependent on tablet dimensions	Size suitable for children	Swallowable	[46]
		Propranolol						
	Tablet	Lumefantrine	Drug content assay	-	Immediate release	Size suitable for children >6 years	Swallowable	[47]
	Oral dosage form	Amiodarone	-	Filament length	-	-	-	[48]
	Tablet	Praziquantel	Drug content assay	Drug concentration	Enhanced drug release	Taste masking	-	[49]
	Tablet	Baclofen	-	Volume	Modified release	Size suitable for children	Swallowable	[50]
	Chewable tablet	Indomethacin	-	-	Immediate release	Candy shapes	Chewable	[51]
						Taste masking		
Drop-on-powder	Tablet	Theophylline	Content uniformity	Volume	Similar to commercial tablet: sustained release (theophylline); immediate release (metoprolol tartrate)	-	-	[52]
				Drug concentration				
		Metoprolol tartrate		Number of ink spraying times				
	Orodispersible tablet	Levetiracetam	-	Volume	Similar to commercial tablet: immediate release	Colorful cartoon shapes	Orodispersible	[53]
Inkjet	Orodispersible film	Warfarin	Content uniformity	Surface area	Immediate release	Acceptable size for children	Orodispersible	[33]
							Administration through feeding tube	
	Orodispersible film	Metoprolol tartrate	Content uniformity	Printing resolution	-	-	Orodispersible	[54]
	Orodispersible film	Salbutamol	Drug content assay	Drug concentration	-	Size suitable for children	Orodispersible	[55]
	Orodispersible film	Clonidine	Drug content assay	Drug concentration	Similar to solvent casting: immediate release	-	Orodispersible	[56]
	Orodispersible film	Enalapril	Drug content assay	-	-	-	Orodispersible	[57]
	Orodispersible film	Vitamin B_1_, B_2_, B_3_, B_6_	Drug content assay	Number of ink spraying times	-	-	Orodispersible	[58]
	Orodispersible film	Propranolol	Drug content assay	Number of ink spraying times	Immediate release	Acceptable palatability	Orodispersible	[59]

## Data Availability

Data available on request. The data presented in this study are available on request from the corresponding author.

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
