# Peer review of "3D Printing of Pediatric Medication: The End of Bad Tasting Oral Liquids?—A Scoping Review"

_pharmaceutics, 2022, doi:10.3390/pharmaceutics14020416_

Round 1
Reviewer 1 Report
Dear authors,
Congratulations on this nice review paper. I have enjoyed reading it, especially the clear overview on clinical trials and advantages compared to conventional manufacturing.
In general, I did however miss sufficient technical insights (see seperate revision document). Please have a look at the suggestions with some suggestions to mitigate/improve this aspect.
Best regards.

Author Response
Dear reviewer,
Thank you very much for the important and thoughtful comments. In the revised version we have addressed all comments and suggestions and believe this has improved the manuscript both with regard to content and readability.
Please find a point-by-point response to the comments in the attachment.

Reviewer 2 Report
In their manuscript “3D printing of pediatric medication: the end of bad tasting oral liquids? – a scoping review” the authors give an overview of current literature on 3d printing with emphasize to a pediatric patient population. The authors used a systematic approach to identify literature, summarize the main literature in form of a table and present results regarding some overlying concepts and disscuss these. Individual studies are not discussed in detail. The article is very well written and conceivable. The title is in my opinion somewhat misleading since the comparison to orally applied liquids is not the focus of this work.
I have the following comments, that I ask the authors to address in their revised version:
- The authors describe their search method. Of 875 records screened, 716 were excluded because they did not meet the inclusion criteria. What are the inclusion criteria?
- Line 315 – 317: “Interestingly, while orodispersible tablets typically had a high drug load, orodispersible films are deemed more suitable for low drug loads.” - The authors seem to be surprised by this. In my opinion this is to be expected because of the low mass of orodispersible films – this would even limit the drug loading capacity if the film is produced with drug in the initial film composition. If drug is printed onto a film substrate, the possibility to achieve a high drug load is further reduced.
- Line 329: “No residual solvents could be detected.” - Was the method used for detection sensitive enough in the authors opinion to provide an estimation of the suitability of the dosage form based on this?
- Line 351 – 353: “… a bioequivalence study in healthy adults has been published (66). They found that the 3D printed tablet was bioequivalent to the reference drug under fasted conditions.” Levetiracetam is a BCS class 1 drug and if I am informed correctly the oral bioavailability is 100 % - therefore, this is not surprising. Might be different for other drugs, even if produced using the same technology.
- Section 3.3.6 target population and disease: I think the reference to the study by Klingmann et al. (84) would also be helpful here in combination with the information how many of the tablets studied would be considered as mini tablets opposed to regular size tablets.
- Line 510 – 511: “However, not much is known on the stability of 3D printed tablets.” In this context especially possible changes regarding drug release due to crystallization of amorphous drug are to be expected for FDM-based dosage forms. Otherwise I would not estimate a much higher risk of instability than for conventionally produced dosage forms.
- Line 514 – 516: “Furthermore, 3D printers have an advantage in the number of manufacturing steps compared to conventional tablet manufacturing. No milling, granulation or compression steps are needed.“ This finding is not necessarily true, at least with the currently available approaches. Prior to filament extrusion, a processing to achieve similar particle size and premixing may also be useful to achieve homogenous filaments. The same should the true for DoP and non-heated extrusion-based techniques. Furthermore, the printing times are momentarily much higher than for conventional tableting – therefore this method may be advantageous for small lots (that might otherwise be compounded) but at least momentarily conventional tableting is much (!) more efficient if the batch size is big enough. This a problem in general – what should we compare 3d printing with? Blockbuster tableting or compounding individual doses?
- Line 550 – 556 Palatability – In the discussion the authors comment on the appropriateness of the approach to make medicines resemble candies. Is there any regulatory advice regarding this?
- Line 619 – 620: “However, FDM usually produces at a high temperature, so thermolabile API’s cannot be processed using this technique.” This is largely dependent on how thermolabile the drug is and what temperatures are needed for filament extrusion and subsequent printing. Different polymers need very different temperatures to achieve sufficient softening to be printed via fdm. There is also some literature on printing of thermolabile drugs via fdm (even though not specifically for pediatrics). Other drugs are acidlabile or prone to hydrolysis, this may also limit processing options. Minor comment: please check the grammar in this sentence (suggestion: FDM usually produces high temperatures, …).
- Lines 629 – 631: “3D printed drug products have proven themselves to be superior in terms of drug dosing and release profile.” Compared to what? A general superiority regarding the release profile is in my opinion not given, and drug dosing in general is certainly not better than in conventional tableting. However, I think the authors are pointing towards individualized dosing and advantages over compounding, which I also consider as strength of 3d printing.
Author Response

(The authors gave the same response as above.)

Reviewer 3 Report
Hello Authors,
The manuscript provides a review about pediatric oral dosage forms using 3D printing technology. Please see my comments below,
"A comprehensive overview of knowledge on the application of 3D printed medicine 94 in the pediatric population is currently missing. Furthermore, there is no roadmap to-95 wards the implementation of 3D printed medicine in the pediatric population" This statement seems overstated. There are several studies about 3D printing for pediatric dosage forms. Some examples are here,
https://www.mdpi.com/1424-8247/14/2/143
https://www.sciencedirect.com/science/article/pii/S1818087621000167
https://www.sciencedirect.com/science/article/abs/pii/S0378517321000053
Author Response
Dear reviewer,
Thank you very much for the important and thoughtful comments. In the revised version we have addressed all comments and suggestions and believe this has improved the manuscript both with regard to content and readability.
Please find our response to the comments in the attachment.
